# The Manifesto of Pharmacoenosis: Merging HIV Pharmacology into Pathocoenosis and Syndemics in Developing Countries

**DOI:** 10.3390/microorganisms9081648

**Published:** 2021-07-31

**Authors:** Mattia Trunfio, Silvia Scabini, Simone Mornese Pinna, Walter Rugge, Chiara Alcantarini, Veronica Pirriatore, Giovanni Di Perri, Stefano Bonora, Barbara Castelnuovo, Andrea Calcagno

**Affiliations:** 1Department of Medical Sciences, Unit of Infectious Diseases, University of Torino, Amedeo di Savoia Hospital, 10149 Torino, Italy; walterrugge@gmail.com (W.R.); chiara.alcantarini@gmail.com (C.A.); veronica.pirriatore@gmail.com (V.P.); giovanni.diperri@unito.it (G.D.P.); stefano.bonora@unito.it (S.B.); andrea.calcagno@unito.it (A.C.); 2Department of Medical Sciences, University of Torino, Città della Salute e della Scienza, 10150 Torino, Italy; silviascabini88@gmail.com (S.S.); simone.mornesepinna@unito.it (S.M.P.); 3Infectious Diseases Institute, College of Health Sciences, Makerere University, Kampala 22418, Uganda; bcastelnuovo@idi.co.ug

**Keywords:** HIV, malaria, tuberculosis, helminths, non-communicable diseases, drug-drug interactions, pathocoenosis, syndemics, pharmacology, pharmacokinetics

## Abstract

Pathocoenosis and syndemics theories have emerged in the last decades meeting the frequent need of better understanding interconnections and reciprocal influences that coexistent communicable and non-communicable diseases play in a specific population. Nevertheless, the attention to pharmacokinetic and pharmacodynamics interactions of co-administered drugs for co-present diseases is to date limitedly paid to alert against detrimental pharmacological combos. Low and middle-income countries are plagued by the highest burden of HIV, tuberculosis, malaria, and helminthiasis, and they are experiencing an alarming rise in non-communicable disorders. In these settings, co-infections and comorbidities are common, but no tailored prescribing nor clinical trials are used to assess and exploit existing opportunities for the simultaneous and potentially synergistic treatment of intertwined diseases. Pharmacoenosis is the set of interactions that take place within a host as well as within a population due to the compresence of two or more diseases and their respective treatments. This framework should pilot integrated health programmes and routine clinical practice to face drug–drug interaction issues, avoiding negative co-administrations but also exploiting potential favourable ones to make the best out of the worst situations; still, to date, guiding data on the latter possibility is limited. Therefore, in this narrative review, we have briefly described both detrimental and favourable physiopathological interactions between HIV and other common co-occurring pathologies (malaria, tuberculosis, helminths, and cardiovascular disorders), and we have presented examples of advantageous potential pharmacological interactions among the drugs prescribed for these diseases from a pharmacokinetics, pharmacodynamics, and pharmacogenetics standpoint.

## 1. Introduction

### 1.1. Pathocoenosis and Syndemic Theories

Historically, infections have been framed as the mere outcome of the relationship between microbes/viruses, human hosts, and the environment that may affect them in terms of infectivity, virulence, transmission, and predisposition. Therefore, the traditional medical approach has long been that of targeting one infectious agent at a time, according to microbiological features and host comorbidities, independently from the wider bio-economic–social context and possible pathogen–pathogen and drug–drug interactions (DDIs). Nevertheless, co-infections are the rule rather than the exception, with several available examples in routine clinical practice [1,2,3,4].

In 1969, the Croatian doctor M.D. Grmek introduced the concept of pathocoenosis to consider diseases of a population as a whole, rather than separately, merging medical, anthropological, historical, and geographical dimensions [5,6]. According to this model, the prevalence and impact of a disease on a given population depends also on the prevalence of all the other co-occurring diseases and on the overall health of the affected population [6].

A further piece of the modern framework was added by the introduction of the concept of syndemic. A syndemic is an interactive aggregation of two or more enmeshed and mutually enhancing diseases that, working together in a context of deleterious social and biological conditions, significantly affects the overall health status of a population [7,8,9]. Deeply influencing and permeating human health, syndemics have been and yet will be a pivotal driver of human history. For the first time, this theory highlighted how health, economic, social, psychological, and anthropological factors promote disease clustering at the population level and impact on disease pathology at the individual level, usually resulting in an increased disease burden and negative effects [7,8,9].

Syndemic interactions may act in several ways and take place at different levels. For instance, even medical treatments can occasionally generate iatrogenic syndemics: Hepatitis B and C infections were spread involuntarily during yellow fever vaccine campaigns and mass intravenous drug administration against schistosomiasis in USA and Egypt, respectively [10,11,12]; besides, drugs to treat one disease may undermine treatments given for other reasons (such as detrimental DDIs between some antiretroviral and anti-tuberculosis drugs).

While in the past, malaria was one of the most important determinants of pathocoenosis, due to its impact on populations’ mortality and morbidity, and to its synergism with other diseases [6,8], HIV infection powerfully emerged in the last century. It has been hypothesised that the initial spread of the HIV pandemic in sub-Saharan Africa was likely eased by industrial evolution as well as by successful vaccine campaigns among tropical countries in the early twentieth century [11,13]. The HIV pandemic is a lifelong process closely entwined with unquantifiable opportunistic and non-opportunistic co-infections, non-communicable diseases, polypharmacy, poverty, marginalisation, gender inequality, malnutrition, and stigma. When populations already affected by adverse socioeconomic and health conditions are exposed to the retroviral “fuel”, the fires of co-existing conditions such as poverty, malaria, or tuberculosis (TB) explode in devastating syndemic interactions [7,9]. This intricate network (of not always direct interactions) is noticeably exemplified by the worrying increase in dog and human rabies incidence in KwaZulu-Natal province of South Africa around 2007, following the high mortality in dog owners from HIV infection, which in turn resulted in rising packs of unvaccinated stray dogs [14]. Luckily, this is not always the case, and even antagonistic interactions can reciprocally interfere among co-infecting pathogens, while biological, psychological, genetic, behavioral, or social characteristics or even diseases can provide protective benefits against other diseases [1,7,8,15]. As a few examples, scrub typhus outbreaks may involuntarily induce the production of potent and long-lasting cross-reactive antibodies able to neutralise CXCR4-HIV-1 viruses [16], and *Trypanosoma cruzi* co-infection might reduce the risk of mother-to-child HIV transmission [17].

Furthermore, counter-syndemics take place when efforts to treat one disease voluntarily or unintendedly help in eliminating another one [7]; campaigns to treat schistosomiasis in Africa would indeed impact also on HIV transmission and disease progression, as endorsed by the association between chronic schistosomiasis, higher risk of HIV acquisition and impaired control on viral replication [18]. As a more recent example, the adherence to health protocols adopted to prevent COVID-19 (for example, self-protective measures such as facial masking) seem to have also reduced the incidence of other endemic or epidemic infectious diseases (influenza, tuberculosis, and pneumonia) [19].

### 1.2. Pharmacoenosis

Pathocoenosis and syndemic theories are not a mere exercise in modelling causalities but may represent useful lenses to identify best practices and further opportunities for the simultaneous and possibly synergistically boosted treatment of intertwined diseases. Then, how could syndemics and pathocoenosis improve our clinical management? Can counter-syndemics be exploited to scale up the global agenda against HIV and co-occurring diseases? Can we turn co-infections and comorbidities into an opportunity for enhancing and expanding the efficacy and effectiveness of vertical programs over simultaneous targets?

To refine the yet unexploited theoretical opportunities of this modern approach, we introduced the concept of pharmacoenosis. By this, we mean the set of interactions that take place within a host as well as within a population due to the compresence of two or more diseases and their respective pharmacological treatments, mediated by pharmacokinetics (PK), pharmacodynamics (PD), and pharmacogenetics (PG) and by host, diseases, and ecological, socioeconomic, and environmental characteristics. A schematic representation of this model is depicted in Figure 1.

DDIs and polypharmacy are usually regarded as an enemy both for patients and clinicians, but they may not be always detrimental, and novel, collateral, unintentional properties of seasoned molecules can be just discovered when administered solely or combined to patients suffering from concomitant diseases [20,21]. It is possible indeed to strategically couple drugs aiming at purposely improving their bioavailability, metabolism, clearance, half-life, and surprisingly even the spectrum of actions [20,21].

Therefore, with this narrative review, we have reported some examples and speculated about possible promising applications of pharmacoenosis among malaria and other communicable and non-communicable diseases commonly co-occurring with HIV in low and middle-income countries (LMIC). Considering that co-infections are eventually worse in terms of health and socioeconomic consequences and that they are usually linked with poorer treatment outcomes and increased costs, especially where health assistance is still the weakest [9], a better understanding of pathocoenosis, syndemics, and pharmacoenosis should be strongly endorsed to pilot integrated programs and routine practices able to face the phenomena and to make the best out of the worst situations.

## 2. Malaria and HIV

HIV and malaria together account for about one million deaths globally every year, the latter being the third cause of HIV-related morbidity where these infections geographically overlap, with an overall co-prevalence of 23–29% and reaching coinfection prevalence of up to 72% in certain sub-Saharan areas [22,23]. While malaria could fuel HIV viral replication [24], Kaposi Sarcoma development [25], sexual and mother-to-child HIV transmission [26], and progression to AIDS [27], HIV has been suggested to increase malaria-associated mortality [28], placental involvement [29], parasite biomass, and the selection of antimalarial resistance [30]. As an example, mathematical models showed that in a small area of Kenya, malaria–HIV co-infection (MHC) has been responsible for 8500 and 980,000 excess in HIV and malaria infections since 1980, with an excess prevalence of 2.1% and 5.1%, respectively [31].

At the opposite of the interactions spectrum, recurrent or prolonged malaria episodes may activate latently infected resting CD4+ T-cells, inducing latent virus reactivation and potentially reducing the reservoir burden in patients on effective combination antiretroviral therapy (cART) [32].

The WHO treatment guidelines for MHC consist of only a few “don’t” rules: to avoid sulfadoxine–pyrimethamine in patients on cotrimoxazole due to the risk of sulfonamide-induced adverse reactions and, if possible, to avoid amodiaquine-containing regimens in patients on zidovudine or efavirenz, as this increases neutropenia, and hepatotoxicity or halofantrine-containing regimens in patients on protease inhibitors, as this would significantly increase the risk of QT prolongation [33]. The scarcity of PK/PD and DDIs studies lead us to caution in several circumstances, such as the co-administration of arthemether/lumefantrine with efavirenz, to the complete lack of recommended associations or preferential strategies (as shown in Figure 2). Yet, encouraging theoretical starting points may be identified.

### 2.1. PD Opportunities

Several protease inhibitors (PIs) and non-nucleoside reverse transcriptase inhibitors (nNRTIs) variably proved advantageous but sparse PD interactions with antimalarial treatments at approaching or at clinically relevant concentrations in in vitro and in vivo animal models: inhibition of *Plasmodium falciparum* growth, enhancement of chloroquine activity and asexual parasites clearance, gametocytocidal and transmission blocking activity, reduction of mosquito infectivity, and gametocyte exflagellation and activity against the hepatic stages of several *Plasmodium* species, including *knowlesi* and *falciparum* [34,35,36,37,38,39,40]. One of the proposed mechanisms for PIs’ chloroquine-resistance reversion has been identified in the reduction of glutathione levels and in chloroquine-resistance-related enzymes [35,38,40]. Despite chloroquine being less used in several LMICs, new chloroquine-like drugs against resistant strains are under development [41]. Considering also the significant structural differences among PI molecules, further studies assessing the antimalarial properties of new-generation PIs should be performed with the perspective of a possible co-administration with regimens based either on chloroquine-like drugs or artemisinin-based combination therapy. Lopinavir proved to advantageously reduce the median effective dose of amodiaquine [39], while indinavir and nelfinavir were proved to enhance artemisinin killing of *Plasmodium falciparum* [42]. However, PIs may also reduce the effect of artemisinin endoperoxides against malaria [36]. Thus, greater clarity should be made upon this issue, as well as upon possible roles of PIs in modifying malaria course and complications due to the double-edged weapon effect of ritonavir and saquinavir in the scavenger receptor CD36 expression (involved in malaria sequestration but also in non-opsonic phagocytosis of parasitised red cells) [43]. Compared to (n)NRTIs-based regimens, PI-based regimens have been associated with lower clinical malaria incidence and risk of recurrent malaria and longer time to first malaria episode [44,45]. Conversely, while chloroquine, primaquine, and mefloquine PD properties against HIV have been already investigated and remain controversial [46], we have not been able to find any data regarding possible antiviral properties of currently used antimalarials, which may be the reflection of either negative-results bias or a real lack of interest about it.

### 2.2. PK and PG Opportunities

Despite antimalarial–antiretroviral interactions represent a frequent risk of toxicity and antimalarial treatment failure, some evidence may suggest tailored PK-based approach. As example, the co-administration of lopinavir/ritonavir with artemether–lumefantrine decrease artemether exposure, but the concurrent increase in lumefantrine exposure has been hypothesised to be one of the reasons behind the lower clinical malaria incidence among patients on PIs-based cART [44,45]. Nevertheless, by inducing a prolonged increase of lumefantrine plasma concentrations through the inhibition of CYP3A4 cytochrome, lopinavir/ritonavir concurrently causes an increased incidence of toxic adverse events among children, while no such complication was reported among adults [47]. These observations suggest the possibility of tailoring treatments according to patients’ genetic background in order to avoid toxicity and improve treatment effectiveness [48].

Indeed, selecting regimen combinations based upon known underlying favourable enzymatic profiles able to compensate in the opposite direction potential unfavourable enzymatic induction/inhibition of DDIs may preserve treatment options, outcomes, and adherence by reducing concentrations below/above the therapeutic range. In this regard, recent data showed a differential impact of the co-administration of nevirapine and antimalarial regimens when stratified by CYP2B6 polymorphisms [49]: the nevirapine-induced reduction of artemether and desbutyl-lumefantrine and increase of dihydroartemisinin and lumefantrine significantly differed between CYP2B6 c516GG versus TT genotypes [49]. If geographical PG data were easily and widely available, population-physiologically-based PK modelling could classify the same co-administration as contraindicated in some areas while recommended or neutral in others. Indeed, the recent availability of dolutegravir in LMICs may overcome some of these issues, as preliminary data reported no relevant clinical reduction of its concentration with standard doses of commonly used antimalarial regimens (as shown in Figure 2) [50].

### 2.3. Companion Drugs

Universal and continuous cotrimoxazole (CTX) use for all people living with HIV (PLWH) in countries with a high prevalence of HIV, malaria, and limited health infrastructure endorsed by the WHO probably represents one of the first clear examples of pharmacoenosis informing public health strategies. The antimalarial activity of CTX is debated: it has been successful in treating uncomplicated *Plasmodium falciparum* infections and, despite not gametocydal, CTX-exposed gametocytes seem to have lower infectivity for *Anopheles*, thereby reducing also mosquito transmission [51]. Several studies demonstrated that CTX maintenance treatment, regardless of CD4 count, in malaria highly endemic countries was associated with a significant reduction in malaria and other bacterial infections incidence, related hospitalisations, and parasitemia burden [52,53]. Nevertheless, some controversy still surrounds this indication, and the benefits should be weighed against higher pill burden, haematological toxicity, adverse reactions, and antimalarial cross-resistance [54]. In this regard, cost-effectiveness and sensitivity analyses could help in delineating tailored settings where long-lasting CTX use can result in the safest and most effective outcomes. 

It has been calculated that the largest epidemiological impact of HIV on malaria (and vice versa) occurs when one disease prevalence is very high and the other is very low and near its endemic threshold [31]. Now that we are approaching a satisfactory malaria control in countries where HIV keeps at an endemic pace, the imbalance in the prevalence of these diseases may represent a peculiar setting rarely occurred before, where the implementation of interventions against HIV may give a further stronger push towards malaria eradication and vice versa. Therefore, more and tailored studies on PD, PK, and PG interactions between antiretrovirals and antimalarials are warranted to lay the foundations of detailed guidelines for groups of patients characterised by specific MHC prevalence and comorbidities who may make the best of them by reducing transmission, severity, and management costs.

## 3. Mycobacteria and HIV

### 3.1. Tuberculosis and HIV

TB is one of the major public health threats worldwide. In 2019, 10 million people developed TB, 8.2% of them were infected with HIV, and an estimated 208,000 TB deaths (range, 177,000–242,000) occurred among PLWH [55]. PLWH have a disproportionate risk of developing active TB compared to HIV-negative individuals, primarily in African countries where three-fourths of new worldwide HIV/*Mycobacterium tuberculosis complex* (MTB) co-infection cases take place [55,56]. The result of this syndemic translates into roughly 208,000 MTB-related deaths among PLWH, representing the first cause of death among them [55]. Indeed, HIV promotes the progression to active TB by disrupting granulomas and increasing the bacillary load, resulting in mycobacterial dissemination and clinically active and infectious TB [56]. On the other hand, mycobacteria can trigger HIV replication in macrophages and T lymphocytes [57]. Co-infected individuals have a greater expression of CCR5 and CXCR4 receptors and larger amounts of pro-inflammatory cytokines that may favour viral replication, persistence, and even HIV myeloid reservoir expansion, accelerating disease progression [58,59,60]. These complex interactions explain why TB incidence has been driven by the HIV epidemic since the 1990s, which is why the co-infection is associated with poorer treatment outcomes and why cART scale-up can lead to a massive reduction in TB incidence [61].

From a pharmacological standpoint, the main issues of HIV-MTB pharmacoenosis are represented by the following: the fact that HIV infection in itself may reduce anti-MTB drugs PK, the higher incidence of anti-TB treatment failures, the many largely acknowledged detrimental DDIs (as shown in Figure 3), and the scarcity of clinical and pharmacological data regarding DDIs between the newest anti-MTB and anti-HIV drugs with possible solutions [62,63].

Despite dolutegravir (DTG) roll-out in LMIC, efavirenz (EFV)-based cART is still among the first antiretrovirals line, and rifampicin strongly decreases EFV plasma levels through CYP2B6 induction [64]. Tropics and subtropics regions retain a disproportionate burden of diseases as well as the most genetic diversity globally, including drug-metabolising and -transporting genes [65]. These areas and primarily Africa are characterised by a large genetic heterogeneity even within the same ethnicity and unique ancestral genetic markers [65]. The CYP2B6 allelic polymorphism 516G > T, highly represented in African people, has demonstrated to increase EFV plasma concentration, reducing virological failure risks but also potentially increasing CNS toxicity [66]. In a Tanzanian study, it has been demonstrated that among subjects with the same CYP2B6 × 6 genotype, EFV clearances were comparable between HIV-infected and HIV/MTB-co-infected subjects regardless of rifampicin administration [67]. The addition of a third player, isoniazid, may modify this CYP2B6-based interaction through the inactivation of CYP2A6 [68]: recent data confirmed the effectiveness of reduced EFV dose when co-administered with rifampicin plus isoniazid [69].

Drawing PG maps where the heterogeneity of genetic polymorphisms involved in drug metabolism is extremely wide could lead to the selection of regimens based on known favourable enzymatic profiles able to compensate in the opposite direction possible negative enzymatic induction or the inhibition of DDIs. Nevertheless, as for tropical infections, pharmacovigilance and pharmacogenetics data in the tropics are neglected; therefore, to date, ARVs dosage adjustments or switch to newly available drugs such as DTG stands as the only “one size fits all” solution to tackle DDIs during anti-MTB treatment.

The increasing incidence of multidrug- (MDR-MTB) and extensively resistant MTB has driven major efforts and interest in the (re-)use of new and old drugs.

Clofazimine, a potent anti-leprosy drug, shows also bactericidal activity against MTB [70]; it has been included as a Group B agent in the updated WHO guidelines for MDR-MTB as well as in shorter regimens also for PLWH without extra-pulmonary MDR-MTB [71]. Although clofazimine should be avoided in co-administration with PIs and rilpivirine due to QT interval prolongation, it is not contraindicated with many other regimens (as shown in Figure 3). Interestingly, a murine model showed that clofazimine enhances the immune response against MTB and the efficacy of Calmette–Guérin vaccine by expanding central memory T-lymphocytes through the blockage of the voltage-gated K+ channel v1.3 [72]. Of note, the HIV protein Tat has been involved in oligodendrocyte and myelin injury through the activation of the same voltage-gated channel [73]. Considering the long duration of clofazimine treatment, potential secondary positive effects of clofazimine against pathogenic mechanisms underlying HIV-associated neurocognitive disorders and HIV-related neurotoxicity should be assessed.

Although no other synergic desirable DDIs between antituberculars and ARVs have been consistently reported, niclosamide has been recently evaluated for its potential dual (anti-HIV/MTB) activity. Niclosamide is an old anti-helminthic drug. It has been shown to affect a number of host signalling pathways including Wnt, Notch, mTOR, NF-kB, and STAT3 [74]. This broad activity has led to its evaluation as a therapeutic agent for several cancers and as a potential antiviral (SARS-CoV-1, Influenza, Chikungunya and Zika) and bactericidal agent (*B. anthracis*, *P. aeruginosa*, and *S. aureus*) [74]. In addition to this, niclosamide seems to possess potential anti-mycobacterial activity being able to inhibit the in vitro growth of MTB strain H37Rv and of *M. abscessus* [75,76]. Recently, Fan et al. proved that niclosamide may both reduce MTB Beijing strain growth and inhibit the replication of integrated HIV-1 in human macrophages and T cells [77]. More interestingly, a simultaneous activity against both HIV and mycobacteria in co-infected human macrophages was observed [77], so that further studies assessing pharmacoenosis benefits in administering niclosamide in co-infected patients are worthy of attention and funding.

### 3.2. Nontuberculous Mycobacteria and HIV

Non-tuberculous mycobacteria (NTM) are ubiquitous opportunists present in soil and water [78]. Data from high-income regions are showing an increasing incidence of NTM infection globally with a large geographic variety per continent and country [78,79]. The lack of epidemiological data from LMIC, rather than being ascribable to different environmental conditions, seems related to competing risks and to the lack of resources supporting the diagnosis. Indeed, the prevalence rate of HIV/NTM co-infection is largely underestimated because not only many cases are misdiagnosed as pulmonary tuberculosis, but also because differentiating between colonisation and disease can be challenging. Considering these, in Africa, the estimates of pulmonary NTM among PLWH ranges from 0.2 to 4%, with a presumptive incidence of 1.8/100 person-year [80,81,82]. Recently, NTM prevalence in pulmonary samples of patients from sub-Saharan Africa was computed as 7.5% [83]: a significant proportion of subjects with NTM had a previous history of pulmonary tuberculosis (32.4%) and were co-infected with HIV (40.5%) [83]; overall, 27.9% of these cases had actually pulmonary NTM disease [83].

Chemoprophylaxis with macrolides such as azithromycin to prevent *M. avium complex* infection had been advocated for HIV patients with CD4 count < 50 cells/µL, but data came from the pre-cART era [84]. Since the advent of cART, the incidence of NTM infections has reduced; thus, the prophylaxis is no longer recommended for patients expected to start cART immediately and could be considered in PLWH not receiving cART or with viral replication despite treatments. Although data on possible PK synergism between ARVs and azithromycin are lacking, there is no contraindication in the co-administration, except for potential QT prolongation with rilpivirine and atazanavir (Figure 3). From a PD point of view, azithromycin has potential efficacy as a chemoprophylaxis and treatment in several infections, including malaria, trachoma, and both endemic and venereal treponematoses [85,86], as well as an immunomodulant with beneficial effects upon morbidity due to HIV-related chronic lung diseases [87]. The opportunities and cost-effectiveness of azithromycin administration should be further detailed in settings characterised by significant incidence of HIV, sexually transmitted infections [88,89], malaria [85], high HIV prevalence among the youngest [90], both elevated mother-to-child HIV transmission and HIV-positive pregnant women [91,92], increasing household air pollution issues [90,93], and low cART adherence/availability [84]. In our opinion, the duration and doses of azithromycin in these settings warrant further randomised clinical studies: the drug is potentially able to indirectly reduce HIV transmission (treating sexually transmitted infections and as chemoprophylaxis against malaria) [85,88,89], to prevent malaria with promising results in pregnancy [91,92] and to act against a plethora of infectious threats for PLWH in LMIC [86,90] at the cost of a very few adverse events but considerable risk of drug resistance development.

## 4. Helminths and HIV

Helminthiasis affects more than a billion people worldwide with a very high prevalence in LMIC [94]. One-third of all soil-transmitted helminthiasis (STH) and more than 90% of cases of schistosomiasis occur in sub-Saharan Africa [95], with significant overlap with the HIV epidemic. Conflicting evidence has been reported about whether and how co-existing helminthiasis can affect HIV transmission and disease progression or, the other way around, whether HIV may predispose to more frequent or severe helminthiasis [96,97]. Nevertheless, treating helminthiasis may actually reduce HIV mother-to-child transmission, HIV viral load, transmission risk, and disease progression, and it improves mucosal immunity and oral vaccine response [98,99]. Antagonist interactions between worms and HIV have also been hypothesised, such as the debated reduced risk for *Strongyloides stercoralis* hyperinfection syndrome in coinfected patients [100].

No data are currently available about combinations that have to be avoided between ARVs and anthelmintic drugs, but possible DDIs should not be expected considering the short course of deworming treatments (see Figure 4).

Ivermectin shows a broad antiparasitic activity and is distributed with mass-drug administrations (MDA) for lymphatic filariasis and onchocerciasis treatment and potentially for malaria control [101]. Ivermectin is also a substrate and modulator of P-glycoprotein/BCRP transporters presenting not yet clear activities on several other efflux proteins, most of which involved in ARVs PK [102,103]. As a result, lopinavir and saquinavir secretory transport seems inhibited by ivermectin, while lopinavir absorption is increased [102,103,104]. While in healthy subjects the latter co-administration could be risky, in helminthic-related malabsorption syndromes, the higher ARVs concentrations induced by ivermectin may reach the therapeutic range during the recovery from malabsorption if properly guided by therapeutic drug monitoring; data on other ARVs are lacking. Moreover, HIV requires importin α/β-mediate nuclear import along with its integrase for productive infection. It has been shown that ivermectin specifically inhibits this pathway in vitro with possible antiviral activity [105]. Considering ivermectin’s long half-life, it may be intriguing to evaluate a potential boosting role of the drug upon the modulation of viral reservoirs in patients on cART.

Another interesting candidate is represented by nitazoxanide, which is a relatively new drug with a broad antiviral and antiparasitic activity spectrum. Nitazoxanide hampers HIV-1 replication in monocyte-derived macrophages by down-regulating CD4 and CCR5 receptors and stimulating antiviral immune responses and the intracellular production of the anti-HIV resistance factors APOBEC3A/3G and tetherin in vitro [106]. Considering its wide spectrum, preliminary evidence of antiviral activity, and recent proposals of three-drug-regimen-based deworming MDA, the possible direct or indirect effects of nitazoxanide-including MDA in HIV high-prevalence areas deserve some attention to potentially lower the pill burden of MDA regimens while showing antiviral activity. Having proved nitazoxanide to possess activity against *Giardia*, *Cryptosporidium*, JCV, HBV, and HCV [107,108], the possible modulation of periodical nitazoxanide administrations upon the transmission and manifestations of these infections should also be assessed.

Lastly, if there is a conceivable a role for anti-helminths in HIV treatment and transmission control, could there also be any potential advantage in certain ARVs regimens against helminthiases? A differential effect of specific ARVs compared to others has been observed on the infection rate and burden per person of helminthiasis, particularly *Trichiuris trichiuria*, among HIV-positive pregnant women [109], but to date, no evident direct or indirect mechanism has been postulated. 

## 5. Non-Communicable Diseases and HIV

Life expectancy lengthening provided by the widespread of cART together with cART-related toxicity, HIV-related chronic inflammation and immune activation, and the rising plague of non-communicable diseases (NCD) in LMIC are leading to a catastrophic unheard-of burden of cardiovascular and metabolic comorbidities (CVMD). The largest portion of NCD burden in LMIC is represented by cardiovascular diseases, followed by cancer, diabetes, and chronic respiratory diseases [110]. NCD account for about 70% of mortality worldwide, of which 67% are in LMIC [110]. This poses new challenges in the long-term clinical management of PLWH, especially in LMIC, where switching to alternative less toxic cART regimens is not always feasible and polypharmacy significantly increases the chance of fueling comorbidities. In this setting, pharmacoenosis becomes essential in reducing drug toxicities and long-term consequences of HIV-related inflammation. 

Hypertension prevalence in sub-Saharan Africa is among the highest and threatens to increase further [111]. Within this framework, hypertension management is complicated by the absence of clear guidelines tailored for local settings and socioeconomic possibilities and frequently the first-line anti-hypertensive treatment consists of a thiazidic diuretic, with specific considerations for diabetes and target organ damage [112], but not for HIV. While thiazide-like diuretics are not recommended in patients with metabolic syndrome because of their adverse metabolic effects, they may be useful in osteoporosis prevention, given their ability to increase bone mineral density by augmenting renal reabsorption of calcium and reducing osteopenic fractures incidence [113]. This feature could be of interest for HIV-infected patients on or coming from TDF-containing regimens.

On the contrary, both angiotensin-converting enzyme inhibitors and angiotensin-II receptor blocker (ARB) have proved to play a protective role in chronic kidney disease progression, being effective in reducing protein excretion in addition to the blood pressure lowering effects. The ARB telmisartan has proved to be capable of inducing a partial but significant reversal of ART-induced metabolic toxicity in vitro models, by modulating genes expression involved in lipid metabolism and by increasing insulin sensitivity [114]. Notably, it has been previously reported that mechanisms either blocking the production or inhibiting the action of angiotensin II, involved in podocyte injury in animal models of HIV-associated nephropathy, can slow the progression of glomerular impairment [115]. In population-based studies, including HIV-positive patients, proteinuria has been also associated with neurocognitive impairment, whose etiology is increasingly attributed to CVMD [116]. Conveniently, telmisartan use has been associated with diminished odds of neurocognitive decline among HIV-negative subjects with baseline macroalbuminuria [116]. To date, the impact of these therapies over the change of neurocognitive function in PLWH is still debated [117]. While ARB may not be suitable as a first-line treatment in LMIC from an economic point of view, if they were proved to affect in the long term the development or the progression of HIV-associated neurocognitive disorders and the ARVs/HIV-related kidney impairment, cost-effectiveness analyses may suggest these drugs as the first-line anti-hypertension drugs in HIV-positive population.

Another example is statins. Immune activation and arterial inflammation induced by HIV infection can persist despite effective cART; therefore, tailored indications for prescribing immunomodulatory drugs such as statins should be investigated. Beyond the traditional lipid-lowering role, statins seem to decrease the plasmatic levels of several markers of inflammation and monocyte/T-cell activation in HIV-positive and negative individuals [118], leading to debated or proved benefits in terms of neurocognition, renal function, and CVMD [119].

Similarly, aspirin has exhibited immune-modulatory properties in HIV-infected subjects, inhibiting platelet activation and blocking pro-inflammatory pathways in several cell lines in vivo [120]. Nevertheless, immune properties of aspirin remain controversial, and its potential applications are under evaluation as an ally to reduce increased cardiovascular risk such as that linked to abacavir-associated platelet hyperreactivity [121].

Potential pharmacoenosis effects should be considered when prescribing drugs in clinical practice, especially in LMIC, where high prevalence of HIV and NCD require an integrated care and a greater awareness of both toxic and beneficial interactions among ARVs and other chronic disease medications.

## 6. Conclusions and Future Perspectives

Current knowledge about multiple diseases interactions is still poor, and randomised controlled trials producing data to guide public health implementation strategies as well as clinical practice usually either do not take into consideration or purposely exclude comorbidities and co-infections. The current pandemic of SARS-CoV-2 entangled with common comorbidities (COPD, diabetes, cardiovascular diseases) and the related chronically prescribed drugs is a clear and tragic example of how the lack of knowledge about pharmacoenosis may lead to chaotic and unreliable drifts of clinical practice. A simplified and schematic representation of DDIs and drug-coinfections among PLWH in LMICs is depicted in Figure 5.

The way we think about diseases affects the way we draw policies and provide care; including pathocoenosis, syndemics, and pharmacoenosis variables when designing randomised clinical trials or in predictive mathematical modelling would be the first step to assess whether and to which extent this approach may impact on diseases control and elimination. Tailored studies would be much more beneficial if the effect of comorbid conditions was investigated to inform physicians on what to do, rather than on what to avoid only. Even the quality and number of evidence supporting the strength of contraindications in co-prescribing should be improved, as one of the reasons of the overall low quality of scientific evidence supporting acknowledged current guidelines may depend on pooling together study populations not properly characterised and defined by comorbidities, co-infections, genetics, demographic, or socio-behavioural features.

Pharmacoenosis could be an opportunity to reduce health expenditures, to scale up the quality, efficacy, and effectiveness of care, and to evaluate the possibility of preserving the highly specific expertise of vertical programmes, meanwhile transforming them into branched programmes with collateral effects, not dispersing energy and resources, but by intertwining each other and being able to create a dense arboreal mantle for a global health agenda in need of renewed oxygen.

## Figures and Tables

**Figure 1 microorganisms-09-01648-f001:**
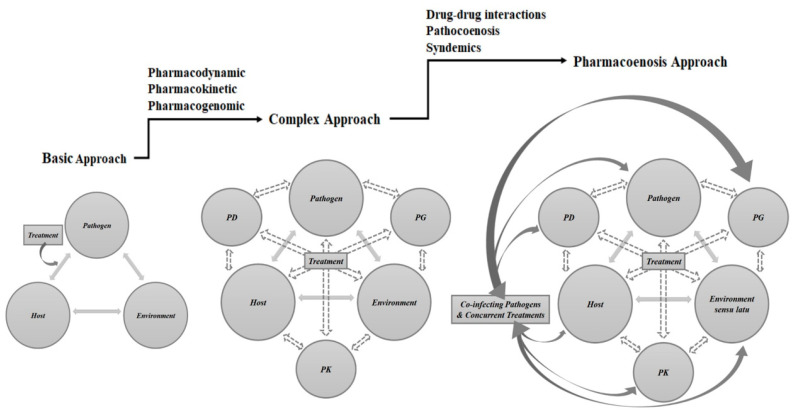
Schematic representation of medical approaches to infectious diseases: the traditional approach, the transitioning approach upgraded by pharmacological considerations that have to be taken into account when prescribing and administering anti-infective drugs and the pharmacoenosis model that tries to tailor the anti-infective treatment based on classical factors (host and pathogen), pharmacological factors, and on potential boosting effects and overlapping benefits of concurrent treatment, infections, and comorbidities. Legend: PD, pharmacodynamics; PG, pharmacogenomics; PK, pharmacokinetics.

**Figure 2 microorganisms-09-01648-f002:**
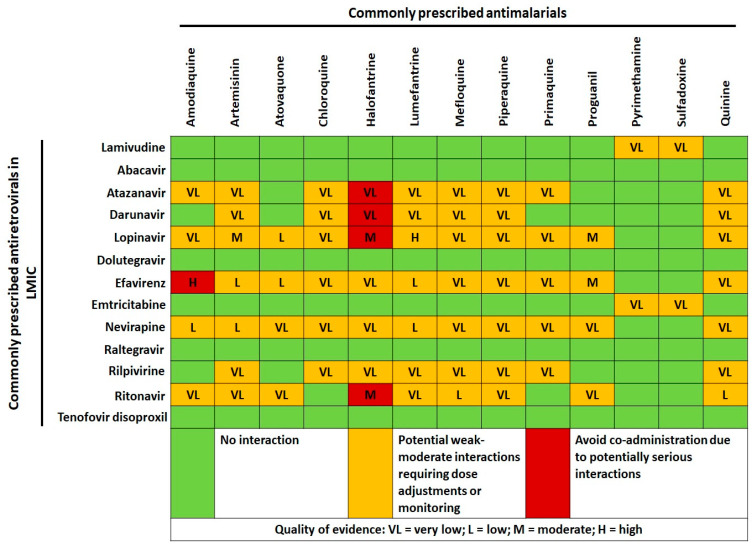
Cross-tabulation of drug–drug interactions between commonly prescribed antimalarials and antiretrovirals in low and middle-income countries (LMIC). Data and quality of scientific evidence were extrapolated from the HIV Drug interaction checker online tool, University of Liverpool (https://www.hiv-druginteractions.org/checker; last access date 20 July 2021).

**Figure 3 microorganisms-09-01648-f003:**
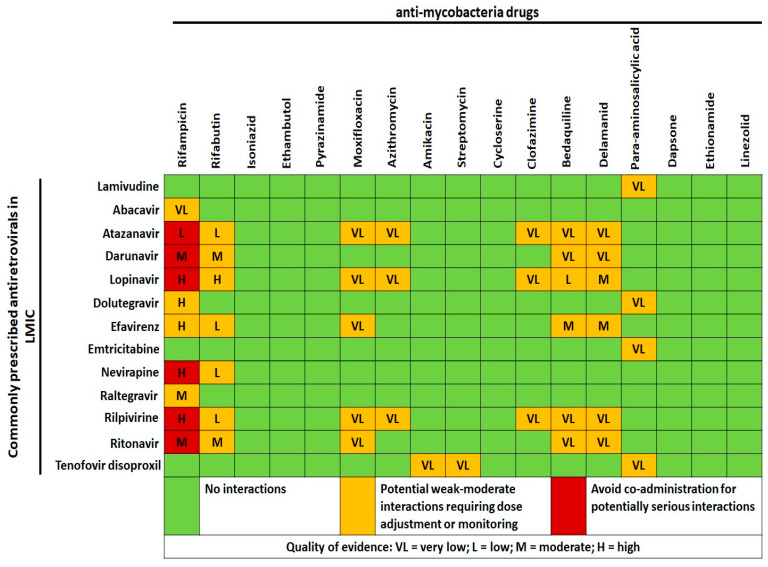
Cross-tabulation of drug–drug interactions between anti-mycobacteria drugs, including newest molecules against multi- and extensively drug-resistant mycobacteria, and antiretrovirals commonly prescribed in low and middle-income countries (LMIC). Data and quality of scientific evidence were extrapolated from the HIV Drug interaction checker online tool, University of Liverpool (https://www.hiv-druginteractions.org/checker; last access date 20 July 2021).

**Figure 4 microorganisms-09-01648-f004:**
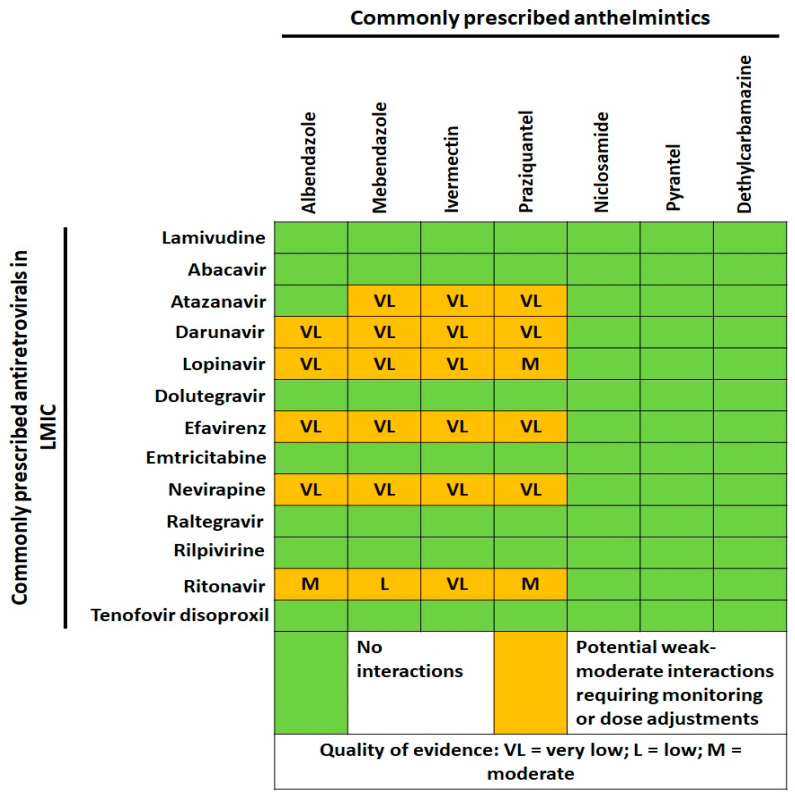
Cross-tabulation of drug–drug interactions between commonly prescribed anthelmintics and antiretrovirals in low and middle-income countries (LMIC). Data and quality of scientific evidence were extrapolated from the HIV Drug interaction checker online tool, University of Liverpool (https://www.hiv-druginteractions.org/checker; last access date 20 July 2021).

**Figure 5 microorganisms-09-01648-f005:**
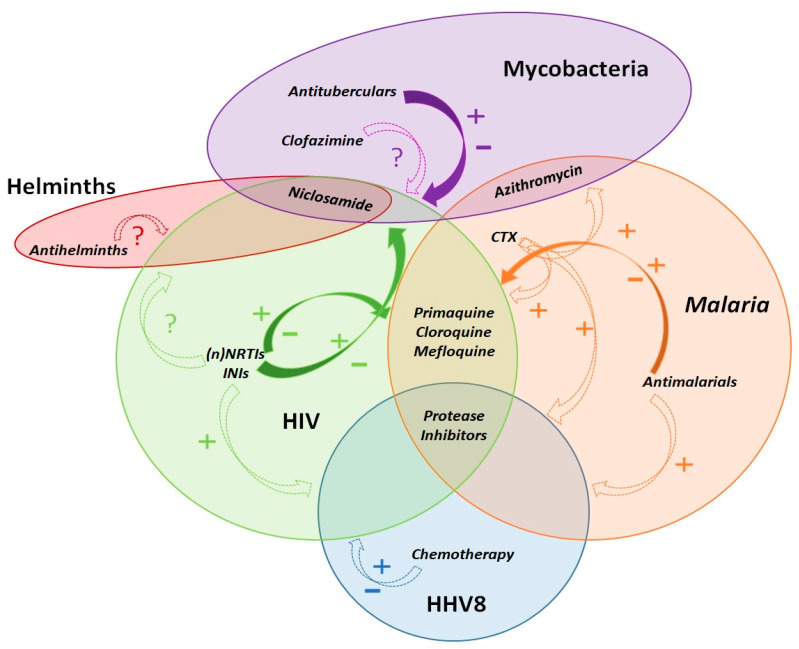
Schematic summary and exemplary representation of observed and uncertain favourable and negative, direct and indirect pharmacological activities of co-administered treatments for HIV and concurrent co-infections. Dashed arrows represent the potential indirect effects of the drug on co-infections, mediated by immunological and/or clinical effects upon their main infective target. Continuous filled arrows represent potential or already known direct effects of the drugs, mediated by pharmacological drug–drug interactions, and they can include also indirect effects through immunological and/or clinical aspects of the disease course. Drugs included within the circles have a direct activity upon the infectious agent causing the specific disease. As an example, some antiretrovirals have known negative and potentially positive direct interactions with drugs prescribed for tuberculosis and malaria; protease inhibitors have also a direct activity upon *Plasmodium* spp. and HHV-8; despite antiretrovirals may not have relevant or common drug-drug interactions with Kaposi sarcoma chemotherapy or anthelmintics, they still have an indirect positive effect upon the evolution of HHV8-related malignancies and helminths co-infections through virological suppression and immunological recovery. Depending on the overlap and co-occurrence of two or more of the depicted diseases, as well as of many others not represented here, this figurative model should lead to prefer a single best therapeutic scheme with the lowest risk of negative drug–drug interactions and the highest potential of boosting or enhancing activities. Legend: +, potential favourable effects; −, potential negative effects; ?, uncertainty about any possible effect; CTX, cotrimoxazole; (n)NRTIs, (non)Nucleoside reverse transcriptase inhibitors; INIs, integrase inhibitors; HHV8, human herpes virus 8.

## Data Availability

Not applicable.

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
