# Peer review of "The Manifesto of Pharmacoenosis: Merging HIV Pharmacology into Pathocoenosis and Syndemics in Developing Countries"

_microorganisms, 2021, doi:10.3390/microorganisms9081648_

Round 1

Reviewer 1 Report

Trunfio M et al. present a captivating review on how the pharmacological strategies used in low and middle-income countries could be improved considering common coinfections. In my opinion this topic is relevant and not sufficiently discussed in the literature.

Major points:

Overall, the authors could improve the review by providing more indications on the “quality” of the existing scientific evidence and being more cautious in reinforcing the theoretical aspect of the proposed model. I consider this aspect to be very relevant to better safeguard the possibility of irresponsible misinterpretation of the work to promote treatments that are not fully validated. Also, some of the references should be updated to include more recent studies.

Other points:     

  • The sentence in lines 14-17 is hard to read and could be rephrased
  • The sentence in lines 21-25 is hard to read and could be rephrased
  • The sentence in lines 59-62 could be rephrased to become more cautious. It should be disclosed that the examples shown represent rare events.
  • Lines 66-68. Are the authors suggesting a synergy between malaria and disasters? It should be removed or clarified.
  • Lines 68-71. Again, in my opinion this sentence could be rephrased to a more cautious formulation. The initial spread of HIV in Africa was a complex and multifactorial process with several unknowns, and this should be highlighted.
  • Lines 88-92. I consider it would be very interesting to very briefly address here the "hot topic" related to the possible impact of COVID-19 preventive measures on other infectious and non-infectious diseases
  • Figure 1: In my opinion the schematic representation created is not very useful or successful in helping to understand the proposed model.
  • Figure 1: The quality of the image is low. Some of the lettering is hard to read.
  • Lines 132-134: Not all of these effects of coinfection are based on evidence with similar degree of confidence. I suggest replacing "can" with "as been suggested to" and add reference to all the research articles presenting the evidence that support each of the statements (viral replication [REF], Kaposi Sarcoma [REF], sexual and mother-to-child HIV transmission [REF]... ).
  • Line 151: I suggest altering “encouraging” to “encouraging theoretical”
  • Figure 2, 3 and 4: The "Quality of evidence" score given by hiv-druginteractions.org should be included in the figure. In example the quality of the evidence supporting that ATV should not be co-administered with Halofantrine is "Very Low", while LPV/r with Halofantrine is "Moderate". This relevant information is easy to include in the figures and is in my opinion very relevant.
  • Figure 2, 3 and 4: for clarification it would be useful to use the complete name of the compounds tested in the interaction checker. In example, instead of Lopinavir write Lopinavir/ritonavir (LPV/r), etc…
  • Figure 5: The figure legend should have more detail to allow a better understanding of the scheme. Also, the resolution and choice of colours could be improved. I don’t see any dashed lines. The yellow lines are hard to see.

Author Response

Trunfio M et al. present a captivating review on how the pharmacological strategies used in low and middle-income countries could be improved considering common coinfections. In my opinion this topic is relevant and not sufficiently discussed in the literature.

AR: Thanks to Reviewer#1 for her/his time spent at improving our manuscript.

Major points:

Overall, the authors could improve the review by providing more indications on the “quality” of the existing scientific evidence and being more cautious in reinforcing the theoretical aspect of the proposed model. I consider this aspect to be very relevant to better safeguard the possibility of irresponsible misinterpretation of the work to promote treatments that are not fully validated. Also, some of the references should be updated to include more recent studies.

AR: Following the suggestions of Reviewer#1 we hope we have better detailed the quality of existing evidence reported in the manuscript now. We have also dampened the strenght of certain sentences regarding our proposed model (see lines 100, 107, 129, 504-508). Lastly, we have updated references reporting epidemiological/incidence/prevalence data of malaria and TB (ref. 23, 51); we are glad to accept specific suggestions for specific references, but unfortunately we have some issues in revising the entire reference list (121 references) to check if there are more recent papers to be reported instead of the one we have used.

Other points:     

  • The sentence in lines 14-17 is hard to read and could be rephrased

AR: Thanks, we have now rephrased the sentence shortening its lenght (see lines 14-17).

  • The sentence in lines 21-25 is hard to read and could be rephrased

AR: Thanks, we have now rephrased the sentence, splitting it into two shorter sentences (see lines 22-27).

  • The sentence in lines 59-62 could be rephrased to become more cautious. It should be disclosed that the examples shown represent rare events.

AR: We have introduced “occasionally” in the sentence to underline the point raised by Reviewer#1 (see line 62).

  • Lines 66-68. Are the authors suggesting a synergy between malaria and disasters? It should be removed or clarified.

AR: We do suggest so; nevertheless, to better explain this concept far more paragraphs would be required. Therefore, being not the main concept we want to convey, we have removed the word “disaster” (see line 70).

  • Lines 68-71. Again, in my opinion this sentence could be rephrased to a more cautious formulation. The initial spread of HIV in Africa was a complex and multifactorial process with several unknowns, and this should be highlighted.

AR: We have modified the sentence accordingly to Reviewer#1’s suggestion, introducing it by “It has been hypothesized that…” (see line 71).

  • Lines 88-92. I consider it would be very interesting to very briefly address here the "hot topic" related to the possible impact of COVID-19 preventive measures on other infectious and non-infectious diseases

AR: We have added a short sentence regarding potential beneficial effect of COVID-19-related protective measures on other infectious diseases (see lines 94-97).

  • Figure 1: In my opinion the schematic representation created is not very useful or successful in helping to understand the proposed model.

AR: We agree with reviewer#1 about the simplicity of Fig.1; nevertheless, we want to use it more like a graphical interlude to break the lenght and potential heaviness of the introduction (readers may be more motivated in keep reading papers presenting figures that break long text). We have expanded the description of fig.1 to better reassume the concept, which is better explained in full in the text (see lines 116-119).

  • Figure 1: The quality of the image is low. Some of the lettering is hard to read.

AR: we have improved the quality of the image.

  • Lines 132-134: Not all of these effects of coinfection are based on evidence with similar degree of confidence. I suggest replacing "can" with "as been suggested to" and add reference to all the research articles presenting the evidence that support each of the statements (viral replication [REF], Kaposi Sarcoma [REF], sexual and mother-to-child HIV transmission [REF]... ).

AR: we have replaced “can” with “has been suggested to” and we have modified the order of the cited references as requested (see lines 141-145)

  • Line 151: I suggest altering “encouraging” to “encouraging theoretical”

AR: done (now line 161).

  • Figure 2, 3 and 4: The "Quality of evidence" score given by hiv-druginteractions.org should be included in the figure. In example the quality of the evidence supporting that ATV should not be co-administered with Halofantrine is "Very Low", while LPV/r with Halofantrine is "Moderate". This relevant information is easy to include in the figures and is in my opinion very relevant.

AR: Thanks, we have improved the messages conveied by fig.2-3-4 showing with a four-degree scale (H, high, M moderate, Lo, low and VL, very low), as suggested, the quality of evidence for potential drug-drug interactions, as reported in hiv-druginteractions.org.

  • Figure 2, 3 and 4: for clarification it would be useful to use the complete name of the compounds tested in the interaction checker. In example, instead of Lopinavir write Lopinavir/ritonavir (LPV/r), etc…

AR: We did not test compounds, but for lopinavir, for which there is no possibility to check the interactions of the single molecule but only of the compound lopinavir/ritonavir. Nevertheless, to report only one drug-one drug interactions (to make the interactions as clearer as possible), for lopinavir we have read the description of each potential interaction reported by hiv-druginteractions.org to assess whether the warning was due to lopinavir or ritonavir (we have also double checked the interactions of ritonavir alone) to report in the figures only those interactions due to lopinavir itself, as for the other molecules.

  • Figure 5: The figure legend should have more detail to allow a better understanding of the scheme. Also, the resolution and choice of colours could be improved. I don’t see any dashed lines. The yellow lines are hard to see.

AR: we have now improved the description of figure 5 (see lines 424-437), the resolution of the figure and the colours choice (removing the yellow spectrum).

Thanks once again to Reviewer#1 for her/his time spent at reading and improving our manuscript.

Reviewer 2 Report

The manuscript submission on The Manifesto of Pharmacoenosis is a very interesting and thought-provoking body of work. It presents concepts which need to be addressed in terms of disease and therapeutic interactions, and how to begin thinking beyond the mere one disease, one treatment approach. The manuscript is well done and I have submitted a few comments to the editor. However I do not have any content suggestions or concerns for the authors.

The manuscript requires some English grammar editing, but minor in nature. Also the figures (outside of heat maps) are of poor resolution and need to have some color modification as well to make them more readable. 

Author Response

Thanks to Reviewer#2 for his/her appreciation of our work.

As highlighted in red in the revised uploaded manuscript we have checked and corrected English language, style and grammar/spelling.

As suggested, we have also uploaded higher resolution figures (1 and 5) with modifications to the colors of figure 5.

Thanks to Rev#2 once again for her/his time spent at reading and improving our manuscript.

Round 2

Reviewer 1 Report

The authors revised the manuscript replying to all raised comments. The new version of the manuscript is much improved.